# Comparative Study of 2D-Cine and 3D-wh Volumetry: Revealing Systemic Error of 2D-Cine Volumetry

**DOI:** 10.3390/diagnostics13203162

**Published:** 2023-10-10

**Authors:** Muhnnad Alkassar, Sophia Engelhardt, Tariq Abu-Tair, Efren Ojeda, Philipp C. Treffer, Michael Weyand, Oliver Rompel

**Affiliations:** 1Department of Cardiac Surgery, Friedrich-Alexander-Universität Erlangen-Nürnberg, 91054 Erlangen, Germany; sophia.engelhardt@web.de (S.E.); michael.weyand@uk-erlangen.de (M.W.); 2Department of Pediatrics, Paracelsus Medical School, General Hospital of Nuremberg, 90419 Nuremberg, Germany; 3Department of Congenital Heart Disease, Centre for Diseases in Childhood and Adolescence, University Medicine Mainz, 55131 Mainz, Germany; tariq.abu-tair@web.de; 4Siemens Healtineers, 91052 Erlangen, Germany; efren.ojeda@siemens-healthineers.com (E.O.); philipp.treffer@siemens-healthineers.com (P.C.T.); 5Institute of Radiology, University Hospital Erlangen, Friedrich-Alexander-Universität Erlangen-Nürnberg, 91054 Erlangen, Germany; oliver.rompel@uk-erlangen.de

**Keywords:** cardiac volumetry, 2D-cine MRI, 3D-wh MRI

## Abstract

This study investigates the crucial factors influencing the end-systolic and end-diastolic volumes in MRI volumetry and their direct effects on the derived functional parameters. Through the simultaneous acquisition of 2D-cine and 3D whole-heart slices in end-diastole and end-systole, we present a novel direct comparison of the volumetric measurements from both methods. A prospective study was conducted with 18 healthy participants. Both 2D-cine and 3D whole-heart sequences were obtained. Despite the differences in the creation of 3D volumes and trigger points, the impact on the LV volume was minimal (134.9 mL ± 16.9 mL vs. 136.6 mL ± 16.6 mL, *p* < 0.01 for end-diastole; 50.6 mL ± 11.0 mL vs. 51.6 mL ± 11.2 mL, *p* = 0.03 for end-systole). In our healthy patient cohort, a systematic underestimation of the end-systolic volume resulted in a significant overestimation of the SV (5.6 mL ± 2.6 mL, *p* < 0.01). The functional calculations from the 3D whole-heart method proved to be highly accurate and correlated well with function measurements from the phase-contrast sequences. Our study is the first to demonstrate the superiority of 3D whole-heart volumetry over 2D-cine volumetry and sheds light on the systematic error inherent in 2D-cine measurements.

## 1. Introduction

The accurate determination of cardiac function is central to cardiac diagnostics. One of the most important aspects is the representation of the pumping performance, i.e., stroke volume (SV). This is strongly dependent on the geometry of the heart in end-diastole and end-systole. Therefore, most imaging techniques for cardiac function measurement are based on volumetric methods that attempt to represent the cardiac geometry over time as accurately as possible. From this, important directly measurable parameters, such as the end-diastolic volume (EDV) and end-systolic volume (ESV), can be determined, from which parameters, such as the ejection fraction (EF) and SV, can be derived.

In the early days of cardiac imaging, it was not possible to present the heart three-dimensionally in its morphology and its geometric changes over time. Hence, estimates were made by measuring the conformational change in a small area of the heart wall and extrapolating to the deformation of the entire ventricle. These measurements usually assumed a cylindrical shape of the left ventricle (LV) [1]. The simplest example of this is the echocardiographic M-mode procedure, where wall movements are determined over time along a beam laid through the heart [2,3]. Given the precise positioning of the beam and the assumption of the cylindrical shape of the heart, the measurement results are valid [4]. However, any deviations from the ideal cylinder shape of the ventricle (in most heart diseases, such as dilation) and imprecise positioning of the measuring beam (examiner-dependent) make this measurement method prone to error [5,6].

Therefore, the ability to fully capture the heart in 2D slice images opened the option for more reproducible volumetry and thus more accurate cardiac function diagnostics. MRI 2D-cine technology has become the gold standard of volumetric analysis [7]. In this imaging technique, the heart is fully acquired by recording 10–20 2D slices in the axial or longitudinal orientation, with 15–25 single images in each slice completely representing the cardiac cycle [8,9]. An ECG-dependent synchronization of the 2D images subsequently allows a 3D reconstruction of the LV at the respective cardiac cycle times. For this, interpolation between the layers is carried out using a cylindrical body representing the blood volume, which is adapted according to the individual circumstances [8,10]. By comparing the relative volume size with the valve opening, it is very easy to determine the EDV and ESV with this method. From this, all cardiac parameters, such as the EF and SV, can subsequently be determined.

In addition to the volumetric approach, it was possible at a very early stage to directly determine the SV by measuring the blood flow velocities, assuming the diameter of the aorta is known. The phase-contrast (PC) measurement carried out for this in MRI is considered extremely valid and not time-consuming, and is therefore the gold standard in determining SV [11,12]. However, an LV volumetry and a statement about the EDV, ES, and EF are not possible with this method, so a valid geometry-based method is necessary for a complete LV function analysis. Additionally, independent studies indicate that LV volumetry using 2D-Cine MRI measurements is susceptible to errors [10,13]. Discrepancies with the measurement results from the PC measurement were shown in repeated studies [14,15]. How this discrepancy arises is not entirely clear. In the past, various reasons have been discussed as triggers. In particular, the lack of inclusion of the papillary muscles [16] and the masking of mitral valve insufficiency were suspected as influencing factors [17].

A crucial point that has not been considered in the previous discussion is the fact that only the incomplete acquisition of the volume is possible through the 2D-cine technique [18,19]. Despite the good approximation of the actual morphology of the LV, the 2D-cine representation is also only an estimate of the three-dimensional shape of the LV based on the summation of the individual 2D layers [20]. The resulting cylinder does not fully take into account the edge structures at the cardiac base and apex (Figure 1c).

In contrast, there is also the possibility of fully capturing the geometry of the heart in three dimensions during a single measurement. Unlike the 2D-cine method, where different individual measurements have to be coordinated using the temporal position in the cardiac cycle, a complete acquisition of the 3D volume is carried out at a certain point in time, which allows a more complete and much more true-to-nature segmentation of the heart [21,22]. The primary application of 3D whole heart (3D-wh) MRI currently lies in the three-dimensional visualization of the heart [23,24]. The acquisition is only performed in one phase of the cardiac cycle, usually in end-diastole. This removes the requirement for interpolation between various slices from different acquisitions, a process that is essential in 2D-cine MRI. Although enhancing the number of slices in 2D-cine MRI might refine the precision, the subsequently formed three-dimensional volume still constitutes a cylindrical reconstruction and is especially susceptible to errors due to motion artifacts.

In isolated studies, it has already been shown that this image acquisition methodology is theoretically also suitable for performing LV volumetry, by creating two images; one in end-systole and one in end-diastole (Figure 1). Despite the theoretical superiority of this technique, a study by Goo et al. [14] could not demonstrate a clear advantage over the 2D-cine technique. While the 3D-wh MRI proved extremely valid in the direct functional comparison with the PC-MRI, discrepancies in determining the volumetry from 2D-cine MRI could not be explained.

The aim of the present work is to uncover causes for the discrepancy in LV volumetry of the new 3D-wh technique compared to the established 2D-Cine MRI procedure, and to evaluate the benefit of 3D-wh MRI for volumetry.

## 2. Materials and Methods

### 2.1. Data Collection

A prospective study was conducted involving 18 healthy students aged 23–25. The intent of the study was solely to refine the measurement of MRI volumetry. The study was conducted in accordance with Good Clinical Practice guidelines. Written consent was obtained from all participants prior to the acquisition of MRI scans. All MRI sequences were performed on each student in a single session. No contrast agent was administered. The study was approved by the hospital’s ethics committee.

### 2.2. Image Acquisition

Measurements were performed on a 3T MRI scanner (MAGNETOM Vida, Siemens, Erlangen, Germany) using a 30-channel phased-array coil.

#### 2.2.1. 2D-Cine MRI

Images were acquired using a balanced steady-state free precession (bSSFP) technique with retrospective ECG gating. Respiratory synchronization was achieved through breath commands given during the end-expiration phase, aligning with our approach for the 3D whole-heart MRI. Depending on heart size, a series of 20–25 short-axis slices perpendicular to the aortic root were acquired. Each 2D short-axis slice represented one cardiac cycle and consisted of 20–25 frames, depending on heart rate. The following settings were used: acquisition matrix 256/146, TR/TE 48.3 ms/1.5 ms, flip angle 38°, turbo field echo factor 9–13, number of cardiac phases 20, temporal resolution 30–40 ms, duration of breath-holding approximately 8 to 12 s, number of signal averages 1, in-plane spatial resolution 1 × 1 mm, 20–25 slices, section thickness 6–8 mm.

#### 2.2.2. 3D-wh MRI

A 3D radial fat-saturated SSFP prototype sequence using self-navigation was performed. This self-navigation technique captures the respiratory motion directly from the ventricular blood pool signal throughout the respiratory cycle, eliminating the need for navigator placement [25]. Accordingly, the image acquisition synchronizes with the end-expiration phase of the respiratory cycle, ensuring data consistency. The settings used were as follows: acquisition matrix 256/512, TR/TE 356 ms/1.5 ms, flip angle 17°, turbo field echo factor 9–13, number of signal averages 1, slice resolution 0.5 mm.

#### 2.2.3. PC MRI

Flow velocity was measured at the ascending aorta using 20 2D-cine PC images acquired during one cardiac cycle. The following settings were used: acquisition matrix 115/192, TR/TE 46 ms/2.5 ms, flip angle 20°, turbo field echo factor 3–4, number of signal averages 2, slice resolution 1 mm.

For accurate alignment of the acquisition over the aorta, two orthogonal balanced steady-state free precession (SSFP) scout images were acquired, perpendicular to each other (coronal and sagittal). The velocity encoding settings were initially set to 150 cm/s, and in the presence of aliasing artifacts, they were increased to 200 cm/s. Analysis was performed using a commercially available workstation (SyngoVia, Siemens Healthineers, Erlangen, Germany).

### 2.3. Quality Assessment of Imaging

The contrast-to-noise ratio (CNR) and signal-to-noise ratio (SNR) were determined via regions of interest (ROIs) using the software syngo.via, version VB60A, accessed on May 2021. The SNR was calculated by dividing the mean signal intensity of the aorta in cross-section (ROI = 1.2 ± 0.1 mm^2^) by the SD of the extra thoracic background noise (ROI = 2.2 ± 0.1 cm^2^) measured from the air surrounding the patient. For the CNR, the mean signal intensity of the LV muscle (ROI = 1.2 ± 0.1 mm^2^) was subtracted from the mean signal intensity of the aorta and afterwards divided by the SD of the background noise.

### 2.4. Image Analysis

#### 2.4.1. Quantification of LV Volumes

##### 2D-Cine

LV volumes were determined from the 2D-cine images using a commercially available workstation (SyngoVia, Siemens Healthineers, Erlangen, Germany). Manual delineation of the endocardium, including the papillary muscles, was performed for each phase (Figure 1). The end-systolic and end-diastolic phases were identified based on the minimum and maximum LV volumes (2D-cine volume). In the case of 2D-cine trigger, the ESV and EDV were determined based on the corresponding trigger time point. The trigger time point for end-diastole and end-systole was obtained from the 3D-wh measurement. The phase closest to these trigger time points was used to determine the EDV and ESV. Heart rates (beats per minute [bpm]) and trigger delays (ms) for 2D short-axis cine imaging and 3D-wh imaging for both end-systolic and end-diastolic phases were obtained from the annotations on the processed images or the Digital Imaging and Communication in Medicine (DICOM) header.

##### 3D Whole-Heart (3D-wh)

3D-wh Volume

For the assessment of LV volumes using the whole-heart sequences, the open-source software 3D-Slicer (Version 5.0.1; http://www.slicer.org, accessed on May 2022) was utilized. The 3D-wh DICOM dataset, acquired in an axial orientation, was imported into the software. Before performing the manual segmentation, the 3D data were converted into short-axis slices using the software’s reformatting tools of 3D-slicer. This orientation facilitates a more straightforward analysis of the LV volumes and mimics the traditional approach used in 2D imaging. Segmentation of the LV volume was then manually executed by setting an optimal threshold, distinguishing the blood pool from the myocardium and vessels, while excluding the papillary muscles. This segmentation extended up to the mitral and aortic valves.

Papillary-3D-wh Volume

Similar to the extraction of the LV volume, manual segmentation of the papillary muscles was performed based on a threshold-based approach.

##### Derived Volumes through Combining 3D-wh and 2D-Cine Volumes

To enable a direct comparison of LV volumes between 2D-cine and 3D-wh, the 2D-cine volumes were transferred to the open-source software 3D Slicer. Using the segmentation workstation, summation calculations were performed based on the volume bodies: 2D-cine, 3D-wh and Papillary-3D-wh volumes. The derived volumes, 2D-cine-lost, 2D-cine-adjust, and 3D-wh-cylinder, were calculated for diastole and systole and visualized in the same 3D coordinate system (Figure 1). The volume 2D-cine-lost represents the volume parts at the base and apex that are not considered in the cylindrical form of the 2D-cine. The volume 2D-cine-adjust corrects the 2D-cine volumes by expanding the 2D-cine lost parts and subtracting the papillary volume. 3D-wh-cylinder was tailored to the cylindrical shape of the 2D-cine for direct comparison of the overlap using the Dice and Hausdorff metrics.

#### 2.4.2. Calculation of Functional Parameters

The functional parameters of EF and SV were calculated. In the case of PC MRI measurements, SV was automatically determined using the corresponding workstation (SyngoVia, Siemens Healthineers, Erlangen, Germany) by integrating the flow velocity in the ascending aorta and the corresponding vessel area. For volumetric measurements, the ESV was subtracted from the EDV. This also provided the EF by dividing the SV by the corresponding EDV.

#### 2.4.3. Degree of 3D Overlap

The two commonly used standard methods, the Dice coefficient and Hausdorff distance, were employed to assess the degree of three-dimensional overlap between the different LV volumes. The Dice coefficient was calculated by summing the overlapping surfaces divided by the average sum of the two surface contours, according to the general equation:$$Dice\;\left(V1,\;V2\right)=2\frac{V1\bar{\cap V2}}{V1+V2}$$

A high Dice’s coefficient value close to 1 indicates a high similarity and strong overlap between the surfaces, while a low value near 0 suggests low similarity or minimal overlap. Multiplying the Dice coefficient by 100 allows expressing it as a percentage of overlap.

The Hausdorff distance measures the average distance between surfaces in three-dimensional space and is reported in millimeters.

These calculations were performed automatically using the SegmentComparison module of the software 3D-slicer (Version 5.0.1; http://www.slicer.org, accessed on May 2022). Both metrics were used to compare the volume of 2D-cine with 3D-wh. To enable a fair comparison, only the cylindrical portion of both volumes was considered. Instead of using the 3D-wh volume, the 3D-wh-cylinder was utilized for the analysis. This approach ensured that only the relevant cylindrical section of the LV was compared between the two volumetric measurements.

### 2.5. Statistical Analysis

Statistical analysis was conducted using SPSS Statistics (Version 21; IBM, Armonk, NY, USA).

The volumes of the LV and individual volume compartments, as well as the SVs, were presented as mean ± standard deviation. Paired *t*-tests were used for comparisons of means. A *p*-value of less than 0.05 was considered statistically significant. The mean differences between the different LV volumes and SV were depicted using Bland–Altman analysis. The mean bias with standard deviation and the 95% confidence interval (mean ± 1.96 SD) were shown. To assess the relationship between the differences in SV and the variations in LV volume measurement, a Pearson’s correlation analysis was performed. A statistical significance level of *p* < 0.05 was considered.

## 3. Results

### 3.1. Image Acquisition Times/Image Quality

The study included 18 subjects with an average age of 23.6 ± 0.6 years, of which 60% were female. The required MRI sequences were successfully performed on all subjects and were free from artifacts. Table 1 demonstrates that both the 2D-cine and the 3D-wh images exhibited above-average image quality metrics. However, the 2D-cine images displayed significantly better contrast intensity (CNR) compared to the 3D-wh images, while the signal quality (SNR) remained comparable (Figure 2).

In terms of the total duration for acquisition and post-processing, 3D-wh imaging required significantly more time compared to 2D-cine (Table 1). This difference was primarily the result of the substantially longer acquisition time for the 3D-wh sequences, which averaged 11.5 ± 7.8 min, in contrast to the mean acquisition time of 3.5 ± 1.6 min for the 2D-cine sequences.

### 3.2. Comparison of Trigger Delay

All subjects exhibited a sinus rhythm without significant frequency variability during the examination, with similar heart rates during the acquisition of both the 2D-cine and 3D-wh images (mean 76.7 ± 11.2 vs. 73.9 ± 9.7, *p* = 0.05). This similarity allows for a direct comparison of the trigger points. Significant differences were observed between the 2D-cine and 3D-wh methods for both the end-diastolic and end-systolic trigger points. In end-systole, the trigger occurred later on average in the 2D-cine imaging compared to the 3D-wh capture (2D-cine: 284.9 ms ± 27.8 ms vs. 3D-wh: 266.9 ms ± 31.7 ms, *p* = 0.02), while in end-diastole, the trigger point in the 2D-cine sequence was earlier on average than in the 3D-wh sequence (2D-cine: 662.5 ms ± 34.9 ms vs. 3D-wh: 684.1 ms ± 39.7 ms, *p* < 0.01).

To investigate the influence of different trigger points on the actual volume of the LV, the ESV and EDV were recalculated from the 2D-cine slices using adjusted trigger points that matched those of the 3D-wh measurement. Subsequently, these 2D-cine-trigger volumes (see chapter Derived Volumes in Section 2) were compared to the regular 2D-cine volumes. Figure 3 displays the volumetric differences in an Altman plot, for better illustration. The altered trigger points had minimal impact on the volumetric measurements of the LV. In both end-diastole (2D-cine: 134.9 mL ± 16.9 mL vs. 2D-cine- trigger: 136.6 mL ± 16.6 mL, *p* < 0.01) and end-systole (2D-cine: 50.6 mL ± 11.0 mL vs. 2D-cine- trigger: 51.6 mL ± 11.2 mL, *p* = 0.03), there was a slight but significant increase in the volume.

### 3.3. Degree of Overlap between 3D Volumes from Cine and Whole-Heart Sequences

Both the 2D-cine and whole-heart procedures involve creating a volumetric representation of the endocardium at end-systole and end-diastole, which is then used to measure the three-dimensional degree of overlap. The comparative analyses are conducted between the 3D-wh-cylinder and the 2D-cine volumes (see Section 2).

Figure 4 illustrates the deviations of the contours between the 2D-cine and 3D-wh volumes. It clearly demonstrates a strong overlap between the two volumes, a finding further supported by the congruence analysis (Table 2). The Dice coefficient reveals an overlap of 90%, with a mean deviation of less than 2 mm. Notably, there is no significant difference in the overlap between diastole and systole.

### 3.4. Volumetric Discrepancies between 2D-Cine and 3D-wh Methods

The comparison between the 3D-wh and 2D-cine volumes revealed significantly higher volumes in both cardiac phases (end-diastole and end-systole) for 3D-wh (Table 3). On average, there were differences of 10.6 mL (±3.8) in favor of the 3D-wh measurements for end-systole and differences of 5.1 mL (±3.1) for end-diastole (Figure 5).

Next, the extent to which differences in the detection of the edge structures (2D-cine-lost) and papillary muscles (papillary-3D-wh) influence volumetry was examined. To address this, the 2D-cine volume was corrected by incorporating the missing edge structures and excluding the papillary muscle components from the LV volume. The comparison with the appropriately adjusted volume 2D-cine-adjust demonstrates a notable alignment between the 2D-cine volumes and the 3D-wh volumes in both end-diastole and end-systole (Figure 5).

### 3.5. Role of Edge Structures and Papillary Muscles in Volumetry: Exploring Their Impact on Volume Measurements

In comparison to the 3D-wh volume, the 2D-cine-generated volumes consistently exhibit a lack of small LV segments from the upper basal and lower apical cardiac structures, which we will refer to as edge structures throughout this analysis (Figure 6c,d). To gain a comprehensive understanding of the influence of these edge structures and papillary muscles on volumetry, each factor was examined individually. Figure 7 highlights the significant differences in the volume between the edge structures during end-diastole (ED) and end-systole (ES). On average, the volume of the edge structure 2D-cine-lost in ES is more than 5 mL larger than in ED (*ES:* 17.5 mL *±* 3.5 mL; *ED:* 12.4 mL *±* 3.7 mL; *p <* 0.01). In contrast, the variations in the papillary muscle volumes between systole and diastole are relatively small, with the volume in systole of the whole-heart sequence being approximately 1 mL smaller compared to diastole (*ES:* 8.5 mL *±* 2.4 mL; *ED:* 9.6 mL *±* 2.5 mL; *p =* 0.04). To investigate the influence of these factors on volume differences, a corrected cine volume (2D-cine-adjust) was constructed, taking into account the adjusted papillary and edge structures for ES and ED. As shown in Table 3, this correction significantly reduces the average deviation from the 3D-wh volume.

A minimal difference in the volume is observed for the papillary muscles between ED and ES. However, a significant disparity in the volume is evident for the edge structures, 2D-cine-lost, with a larger volume in end-systole compared to end-diastole.

### 3.6. Comparison of Heart Function Measurements

The aim of this part of the study was to assess the accuracy of the cardiac function measurements derived from 3D-wh and 2D-cine MRI by comparing them to the gold standard method of PC.

Table 4 displays the comparison of SV values. The 2D-cine measurements demonstrated significantly higher SV values compared to the PC standard (2D-cine: 84.4 mL ± 11.4 mL vs. *PC:* 78.8 mL ± 12.0 mL*; p* < 0.01, mean difference 5.6 ± 2.5 mL). However, there was no significant difference between the SVs determined by the 3D-wh measurements and the results obtained from the PC measurements (3D-wh: 79.0 mL ± 12.1 mL vs. *PC:* 78.8 mL ± 12.0 mL; *p =* 0.73, mean difference 5.6 ± 2.5 mL). When incorporating the edge structures and papillary muscles, the volumetry correction for 2D-cine volumes resulted in a closer approximation to the SV obtained from the PC measurements (2D-cine-adj: 78.2 mL ± 12.0 mL vs. *PC:* 78.8 mL ± 12.0 mL; *p =* 0.73). Figure 8 illustrates the deviations of volumes in an Altman plot, indicating a mean deviation of +5mL per stroke for SV measurements using the cine method.

To assess the influence of the edge structures and papillary muscles on the difference in SV between the 2D-cine SV and PC SV, a correlation analysis was conducted. The results revealed a strong correlation between the differences in the SV and the difference in the residual volumes between end-diastole and end-systole (Figure 9). A higher difference in the residual volumes between ED and ES corresponded to increased deviations in the SV. However, no significant correlation was observed for the papillary muscle differences.

## 4. Discussion

Advancements in medical imaging technology consistently provide novel opportunities for cardiac volumetry. Currently, 2D-cine imaging holds the clinical gold standard for the determination of cardiac functional parameters. More recent studies have increasingly deployed the 3D-wh technique for cardiac volume measurement [21,24], due to its capability to provide a detailed reconstruction and comprehensive representation of the cardiac geometry. However, the existing literature has yet to unequivocally establish the superiority of the 3D-wh approach over the 2D-cine method.

Consistent with Goo’s study [14], a discrepancy was also observed in this work between the 2D-cine and 3D-wh methodologies when assessing LV volumetry. Given that the cardiac function values derived from these measurements significantly influence clinical practice, obtaining the most precise measurements possible is crucial for accurate diagnosis, therapy selection, surgical indication, and prognosis [26,27]. Accurate volumetric methods are particularly vital in cases of pre-existing heart pathologies, such as congenital heart defects or cardiomyopathies [27].

### 4.1. Quality Assessment

The results in Table 1 demonstrate a remarkably high image quality in both datasets, with excellent blood–myocardium contrast. This is evident from the corresponding contrast-to-noise ratios and signal-to-noise ratios. It was observed that both the average acquisition time (in minutes) and subsequent segmentation time (in minutes) for ventricle reconstruction were lower in the 2D-cine approach compared to the utilization of the 3D whole-heart technique. We opted for manual endocardial delineation as it remains the most reliable method for complex heart defects and specific pathologies. This method, despite being manual for 2D-cine volumetry, still takes considerably less time compared to 3D-wh volumetry. The automated processing of 2D-cine volumetry, in non-congenital cases, can often be completed in under a minute.

### 4.2. Trigger Time Discrepancies in 2D-Cine vs. 3D Whole-Heart

Both the 2D-cine and 3D-wh recordings were performed sequentially for each patient, with no significant deviations in the heart rate observed between the two recordings. This allowed for a direct comparison of the recording times based on the registered trigger times. Consistent with the findings of Goo et al. [14], our measurements also revealed significant timing differences between the 2D-cine and 3D-wh recordings, both in end-systole and end-diastole. In both cases, the timing for the 3D-wh recording was slightly delayed compared to the 2D-cine recording. Our measurements also showed that the end-systolic trigger times for the 3D-wh recordings were slightly longer than those of 2D-cine recordings. Conversely, our end-diastolic measurements of the 3D-wh recordings were significantly longer than those of the 2D-cine recordings. In contrast, Goo et al. reported slightly shortened trigger times during end-diastole of the 3D-wh recordings. This led them to conclude that an overestimation of the ESV and an underestimation of the EDV must occur, thus attempting to explain the volume differences between the two methods. To test this assumption, we calculated the 2D-cine volumes equivalent to the trigger times of the 3D-wh sequences for both end-systole and end-diastole and compared them with the original 2D-cine volumes in ED and ES.

As shown in Figure 3, these changes were minimal in both end-diastole and end-systole, and although there were significant volume deviations, they were negligible compared to the magnitude of the discrepancies between 3D-wh and 2D-cine (Figure 5). Therefore, they cannot account for the resulting discrepancies in the functional measurements. The timing of the trigger in end-systole and end-diastole, influenced by the employed technique, plays a role here. We utilized respiratory correction, which enabled the recording of good-quality end-systolic trigger times. The average difference in the volume was approximately 1–2 mL in both end-systole and end-diastole. These findings align with previous observations of minimal volumetric changes around the end-systole and end-diastole phases [28]. Supporting this, a high degree of overlap was observed in the direct three-dimensional comparison of the endocardial inner volume between 3D-wh and 2D-cine (Figure 3).

In summary, it can be concluded that minor deviations in the trigger time around the end-systole and end-diastole phases have only a marginal influence on volumetric measurements.

### 4.3. Reconstruction of Blood Volume from Various 2D Images in 2D-Cine Volumetry—How Large Are the Deviations When Directly Compared with Whole Heart Volumetry?

The method of reconstructing the EDV and ESV fundamentally differs between the 2D-cine and the 3D-wh procedures [29]. In the 2D-cine process, 15–25 layer images are generated during a cardiac cycle, with a predetermined spatial region set for measurement (Figure 1). Reading from the k-space results in a planar depiction of the heart’s contraction, with the read-out area content representing a fixed region during the cardiac cycle. The full cardiac image is obtained through the generation of several such 2D-cine recordings, usually in an axial direction to the heart axis. By segmenting the blood–tissue boundary, the area of the blood portion of a layer at a specific time can be calculated. This area is then multiplied by the layer thickness to obtain a volume of the inner lumen for each layer [30]. By summing all the recorded layer volumes at a specific time, the total volume can be calculated and represented three-dimensionally as a cylinder for various points in the cardiac cycle. In this process, the generation of the individual heart layers is independent, and a suitable interpolation of the layer boundaries is required to generate a total volume. Movement artifacts caused by breathing or irregular heart contractions or arrhythmias complicate the generation of a real volumetric heart cylinder due to shifts of individual planar recordings in space, thus necessitating good interpolation [31].

In contrast, the 3D-wh method does not rely on interpolation; the data capture of the entire heart occurs in one measurement, each at a precisely defined trigger time, in ED or ES [32]. Here, too, movement artifacts negatively affect image acquisition, but artifacts lead to a deterioration of image quality and can be measured by SNR and SN. As visible in Table 1, our measurements had an above-average good CNR and SNR, so it can be assumed that there were few artifacts and an accurate representation of the heart at the recorded phase. Interestingly, our study revealed that, despite the fundamental differences in image acquisition between the 2D-cine technique (with a significantly lower slice resolution) and the 3D whole-heart technique, there’s a high overlap of the 3D volumetrics (Table 2). Consequently, the derived comparative volume from both methods is almost identical (Table 3).

#### 4.3.1. Influence of Papillary Muscles on LV Volume

In various studies, the influence of the papillary muscles on volumetry has been investigated. Similar to our findings, most studies show significant differences in the EDV and ESV depending on whether the papillary muscles are included or excluded from the blood volume during 2D-cine reconstruction. The inclusion of the papillary muscles in the LV blood volume and their exclusion from the blood volume are common procedures, leading to the inconsistent handling of measurements [16,33,34,35].

Despite the demonstrated influence on the calculated blood volume in end-systole and end-diastole, most studies show only a minor impact on functional parameters such as the EF and SV [36,37]. Even in certain conditions, such as dilated cardiomyopathy, the influence is relatively small [37]. In contrast, the inclusion of the papillary muscle volume in the blood volume significantly affects the EF in hearts with hypertrophic cardiomyopathy [38]. However, unlike in healthy individuals, the weight of the papillary muscles is significantly increased in hearts with hypertrophic cardiomyopathy, accompanied by a decreased EDV and ESV [39]. Interestingly, this population also does not show a significant change in the SV. This can be understood by comparing the weight of the papillary muscles in ED and ES. In most studies, there is no significant difference in the absolute papillary muscle volume between ED and ES [36,38]. The increase in the blood volume in ES and ED by the same amount results in no change in the SV. However, the EF is influenced depending on the difference in the papillary muscle volume between ED and ES. For most healthy individuals and many diseases, this only leads to a minor impact on the EF [37], whereas in the case of hypertrophic cardiomyopathy, the significant increase in the difference in the papillary muscle volume between ED and ES has a greater influence on the EF [38].

Consistent with our results in healthy subjects, the consideration of papillary muscle volumes has a negligible impact on the SV. In our measurements, the papillary muscle volumes in ES and ED were comparable (Figure 7), with a mean difference of 1.2 (±0.8) mL. Due to the relatively small volume contribution in ES and ED, the influence on the EF is also minimal.

#### 4.3.2. Influence of Incomplete Detection of Structures near the Base on Volumetry and Role in Functional Assessment

One particular challenge in volumetry using 2D-cine is the inaccurate capture of the basal and apical edge structures. Early on, it was recognized that incomplete acquisition, especially of the basal structures, occurs during short-axis 2D-cine imaging [18]. This is primarily due to the up-and-down motion of the heart along the cardiac axis during a cardiac cycle [40], while the cine slices are acquired in fixed spatial positions. Figure 1 illustrates the process of 2D-cine acquisition. The first 2D slice is positioned at the level of the AV plane during end-diastole, while the last slice covers the heart apex, also during end-diastole. Due to the downward motion of the heart, the first slice plane in systole does not correspond to the first slice plane in diastole and often only includes atrial lumen. Various factors, such as the extent of the longitudinal contraction and axial myocardial shortening, influence the extent of the downward motion, resulting in different slice planes in systole that may be intraventricular. These planes need to be individually determined.

Given this knowledge, the significant deviations in the ESV from those of the 3D-wh measurements are understandable. Since in many slices, the first possible slice is still located just above the AV plane within the atrium, we are forced to use the next slice as the first slice for LV cylinder reconstruction, which results in a portion of the LV volume at the heart base not being considered (Figure 1). This leads to a systemic error in our measurements, with a significantly larger LV volume in systole that is not included in the volumetry, resulting in an underestimation of the ESV (Table 3). Additionally, with the 2D-cine method, the patient-specific features at the AV valve and aorta cannot be imaged, as the 2D-cine volumes represent cylindrical bodies with a flat surface. Congenital heart defects, such as AV canal disorders where the AV valve and aortic valve cannot be visualized in the same plane, also inherently include a systemic measurement error.

Constructing corrected 2D-cine cylinders (2D-cine-adj) in ES and ED including the edge structures resulted in a volume alignment when directly compared to the 2D-cine volumes in both ED and ES (Figure 5, Table 3). This demonstrates the significant impact of inadequate acquisition of the edge structures on 2D-cine volumetry.

Although the issue has long been recognized, to our knowledge, this is the first study to demonstrate this systemic error in a study involving healthy subjects.

To achieve a better approximation of the LV volume to reality, there have been attempts to correct the created cine cylinders for a long time. One of the initial methods addressing the inaccuracy of systolic volumes involved including the slices predominantly created in the atrium and subsequently subtracting the atrial volumes [13]. More recent studies have incorporated 2D long-axis slices (LAX) in addition to the 2D short-axis (SAX) slices for automated ventricle reconstruction [41,42], simulating the ESV and EDV. Xia et al. developed a deep learning-based algorithm that suggests automatic completion of missing structures in the reconstruction [19,43].

### 4.4. Influence of Factors on Cardiac Functional Parameters: Implications for Volumetric Measurements

When comparing the calculated SVs of both volumetric measurement methods with the gold standard of cardiac functional measurement, PC MRI, a consistent overestimation of the SV parameter is observed in 2D-cine derived measurements (Table 4). In contrast, the SV derived from 3D-wh volumetry shows no significant difference compared to PC MRI.

Accurate anatomical reconstruction of the true ventricular inner lumens at the respective time points is crucial for calculating the SV from the ESV and EDV. Various factors, such as reconstruction errors in the 3D volume in 2D-cine methods, the inclusion or exclusion of the papillary muscles in the blood volume fraction, and incomplete volumes due to missing edge structures directly affect the EDV and ESV and subsequently influence the calculated functional parameters. Despite the awareness of these factors, there is a lack of studies systematically analyzing their impact on 2D-cine volumetry.

To address these issues, we performed corrections on the volumes derived from 2D-cine slices by accounting for missing regions resulting from heart motion and inclusion of the papillary muscles using the whole-heart slices. The resulting functional parameters after this correction aligned well with the measurements from PC MRI (Figure 8).

Similar to our findings, many other research groups have also reported a significant overestimation of functional parameters, such as the EF and SV, from 2D-cine slices compared to PC measurements. These results contradict the intuitive expectation of underestimation due to the smaller cylindrical shape. However, no study has provided a conclusive explanation for this discrepancy based on measurements. Speculations have been made regarding volume loss through myocardial infarction or the influence of included papillary muscles.

In this study, we demonstrate for the first time that the crucial effect leading to the observed discrepancies lies in the incomplete ventricle capture during the 2D-cine method, with differences in the manifestation between systole and diastole. Our results show a systematic underestimation of the ESV, resulting in an increase in the derived functional parameters (SV and EF). This finding is further supported by a strong linear correlation between the differences in volume changes and the differences in the SV compared to the gold standard. Interestingly, no such correlation is observed for the papillary muscles (Figure 9).

## 5. Conclusions

In this study, we systematically compared 2D-cine and 3D-wh MRI volumetry, advancing our understanding of the differences between these two methods. We found that discrepancies between the methods were not significantly influenced by the creation of 3D volumes or trigger time points, but were impacted by the inclusion of the papillary musculature and the incomplete acquisition of the edge structures in 2D-cine due to heart motion. In direct comparison with the PK measurements, deviations in the papillary muscle volume minimally affected the SV; however, the systematic underestimation of the ESV due to the incomplete acquisition of edge structures in 2D-cine led to a significant overestimation of the SV. This systematic error may be more pronounced in pathologies affecting heart morphology or longitudinal movement. Automated correction strategies for 2D-cine volumes, including landmark recognition or deep learning, have potential to improve the accuracy of 2D-cine volumetry but are rarely used in clinical practice. The accuracy of these methods remains insufficiently verified.

In conclusion, based on our study’s findings, we believe that the application of 3D whole-heart volumetry is superior to the current clinical usage of 2D-cine volumetry.

## 6. Limitations

Regarding limitations, our study’s patient cohort is relatively small, potentially affecting the statistical power and generalizability of our findings. Additionally, the cohort comprises only healthy individuals, raising uncertainty about the transferability of the results to patients with cardiac pathologies, particularly those with regional wall motion abnormalities. Another point to consider is the inaccuracy of PC measurement when taken at the ascending aorta. Various studies have demonstrated that measurements taken directly at the valvular level, especially in heart pathologies, can lead to an underestimation of volume [44,45]. Such inaccuracies must be taken into account when designing new studies that aim to include heart pathologies. Future investigations with larger and more diverse patient cohorts are essential to further validate the applicability and accuracy of 3D whole-heart volumetry in various clinical scenarios.

## Figures and Tables

**Figure 1 diagnostics-13-03162-f001:**
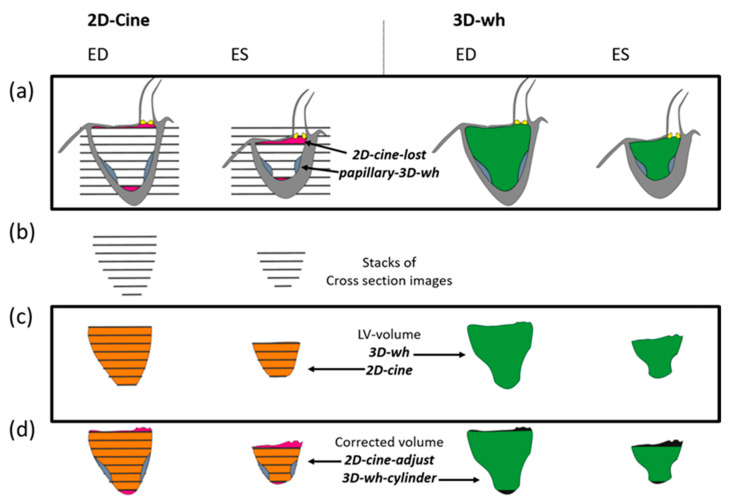
Approach for generating LV volumes from 2D-cine and 3D-wh sequences. (**a**) Generation of LV volumes in 2D-cine: acquiring stacks of cross-sectional images at different phases of the cardiac cycle. Due to heart motion, cardiac base and apex are not captured (2D-cine-lost). In contrast, 3D-wh imaging acquires the entire heart volume. (**b**) The individual cross-sectional images are synchronized according to the cardiac phase, representing end-systole and end-diastole. (**c**) Interpolation leads to the creation of a combined cylindrical volume in 2D-cine, where the volume of the papillary muscles (papillary-3D-wh) is included. The 3D-wh segmented volume, on the other hand, is comprehensive and does not contain papillary muscle volume. (**d**) Construction of corrected volumes: 2D-cine adjust = 2D-cine—papillary-3D-wh + 2D-cine lost, 3D-wh-cylinder = 3D-wh—2D-cine lost.

**Figure 2 diagnostics-13-03162-f002:**
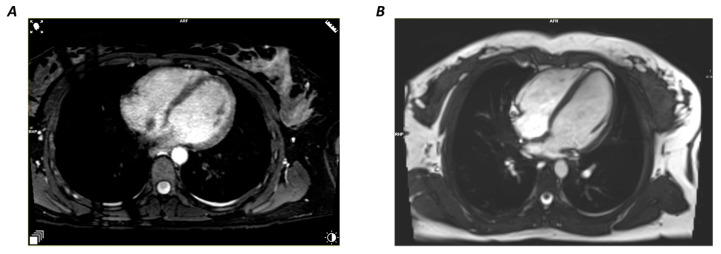
Comparison of the Four-Chamber View: Both 2D-cine (**A**) and 3D-wh (**B**) demonstrate high CNR and SNR, with a slight advantage in CNR for the 2D-Cine.

**Figure 3 diagnostics-13-03162-f003:**
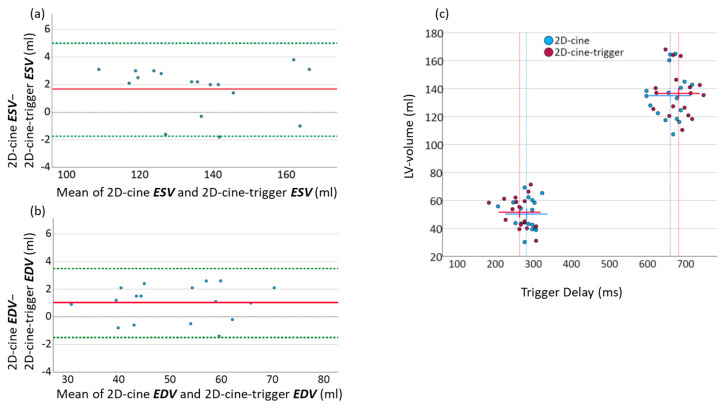
Bland–Altman plots illustrating the mean deviation between volumes generated using 2D-cine MRI and 3D whole-heart MRI. Additionally, the mean deviation after appropriate correction of the 2D-cine volumes is presented. The red solid line in subfigures (**a**,**b**) represents the mean deviation, and the green dotted line indicates the 95% confidence interval. (**a**) Bland–Altman plot displaying the mean deviation in end systole between 2D-cine and 3D-wh volumes (**b**) Bland–Altman plot displaying the mean deviation in end diastole between 2D-cine and adjusted 2D-cine volumes. (**c**) Impact of triggering delay on volume in end-systole and end-diastole. Individual values are represented as points, with bars depicting the mean values. The original 2D-cine trigger points are shown in blue, while the cine time points adjusted to the whole-heart delays (2D-cine-trigger) are presented in red.

**Figure 4 diagnostics-13-03162-f004:**
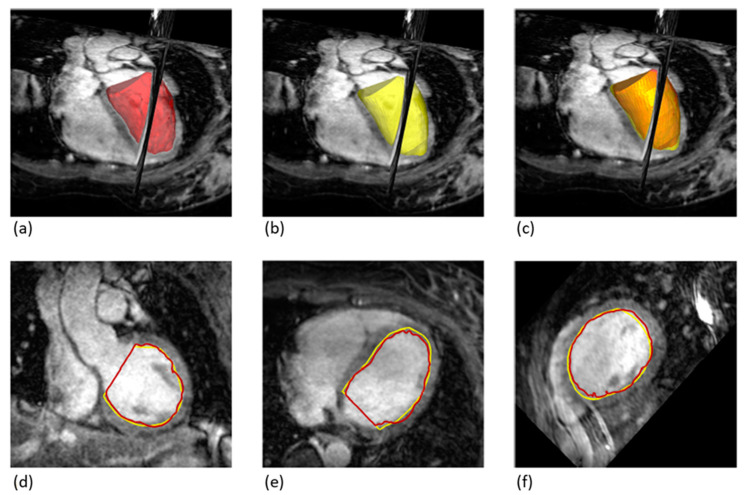
Comparison of Overlapping Volumes between 2D-cine and 3D-wh Techniques. (**a**) 3D Volume Reconstruction using 3D-wh (red), showing a reconstructed cylinder for direct comparison (3D-wh-cylinder). (**b**) 3D Volume Reconstruction using 2D-cine (yellow). (**c**) Merge of A and B, illustrating the degree of overlap between the 3D-wh and 2D-cine volumes. (**d**–**f**). Endocardial Overlapping in Different Slice Planes.

**Figure 5 diagnostics-13-03162-f005:**
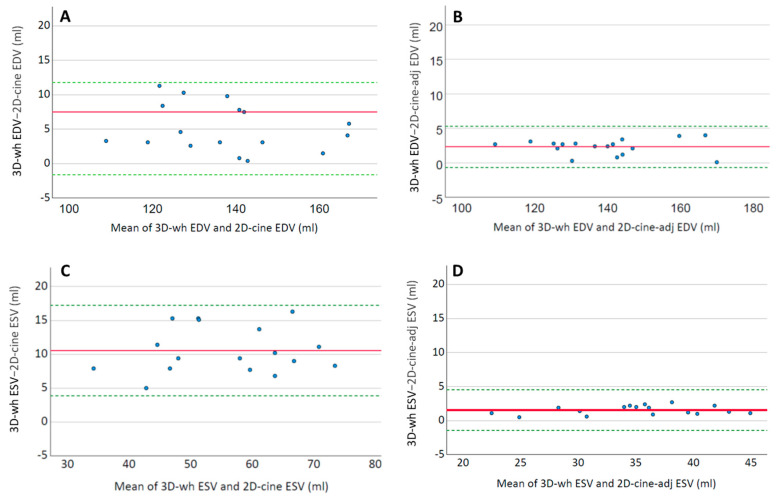
Bland–Altman plots illustrating the mean deviation between volumes generated using 2D-cine MRI and 3D whole-heart MRI (**A**,**C**). Additionally, the mean deviation after appropriate correction of the 2D-cine volumes is presented (**B**,**D**). The red solid line in subfigures (**A**–**D**) represents the mean deviation, and the green dotted line indicates the 95% confidence interval. (**A**) Bland–Altman plot depicting the mean deviation in end-diastole (ED) between 2D-cine and 3D whole-heart MRI. (**B**) Bland–Altman plot displaying the mean deviation in ED between 2D-cine and adjusted 2D-cine volumes (**C**) Bland–Altman plot illustrating the mean deviation in end-systole (ES) between 2D-cine and 3D whole-heart MRI. (**D**) Bland–Altman plot showing the mean deviation in ES between 2D-cine and adjusted 2D-cine volumes.

**Figure 6 diagnostics-13-03162-f006:**
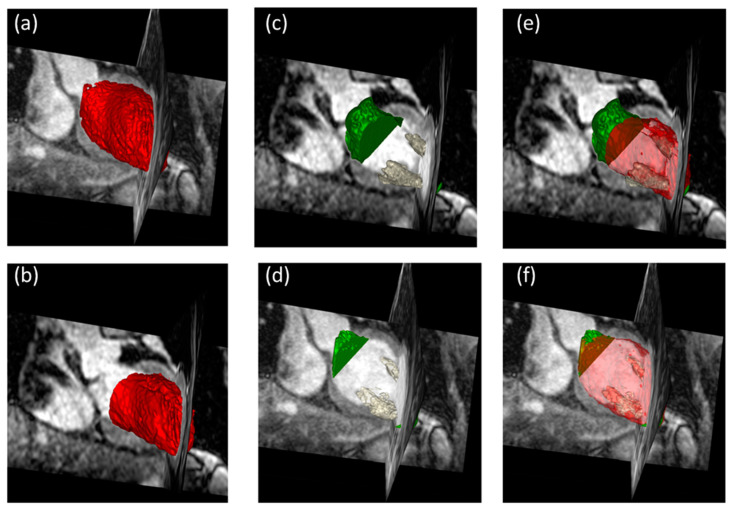
Three-dimensional visualization of segmented LV volumes in both ES and ED. (**a**) 3D-wh total volume in ED (colored in red). (**b**) 3D-wh total volume in ES (colored in red). (**c**) Apical and basal edge structures (highlighted in green as 2D-cine-lost) and papillary muscles (colored in white) visible in 3D whole-heart but absent in 2D-cine in ED. (**d**) Apical and basal edge structures (highlighted in green as 2D-cine-lost) and papillary muscles (colored in white) visible in 3D whole-heart but absent in 2D-cine in ES. (**e**) 2D-cine-adjust, obtained by adding 2D-cine-lost to 2D-cine and subtracting papillary muscles in ED. (**f**) 2D-cine-adjust, obtained by adding 2D-cine-lost to 2D-cine and subtracting papillary muscles in ES.

**Figure 7 diagnostics-13-03162-f007:**
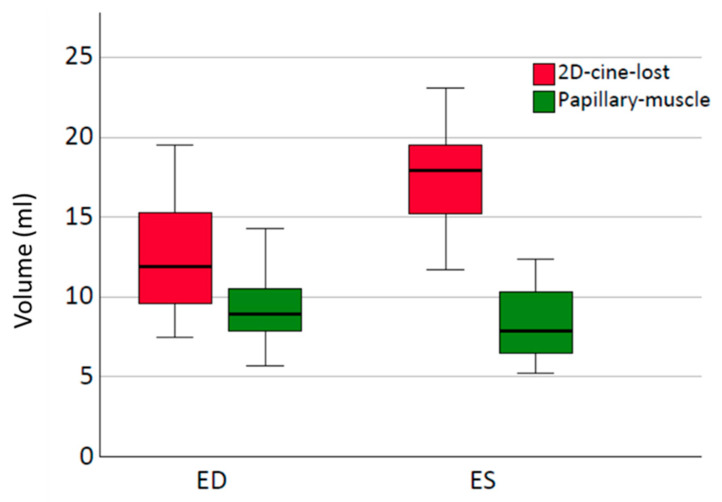
Representation of volumes of papillary muscles and remaining structures in end-systole (ES) and end-diastole (ED) using a Boxplot, showing the median with 25% and 75% quartiles.

**Figure 8 diagnostics-13-03162-f008:**
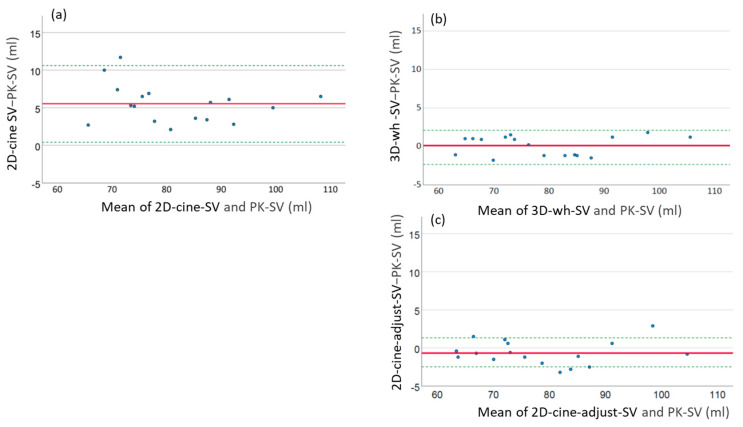
Bland–Altman plot illustrating discrepancies between SV from PC (PK) measurement and volumetric assessments. The red solid line in subfigures (**a**,**b**) represents the mean deviation, and the green dotted line indicates the 95% confidence interval. (**a**) Bland–Altman plot showing the mean deviation of SV between 2D-cine and PK measurement. (**b**) Bland–Altman plot displaying the mean deviation of SV between 3D-wh and PK measurement. (**c**) Bland–Altman plot indicating the mean deviation of SV between corrected 2D-cine (2D-cine-adjust) and PK measurement.

**Figure 9 diagnostics-13-03162-f009:**
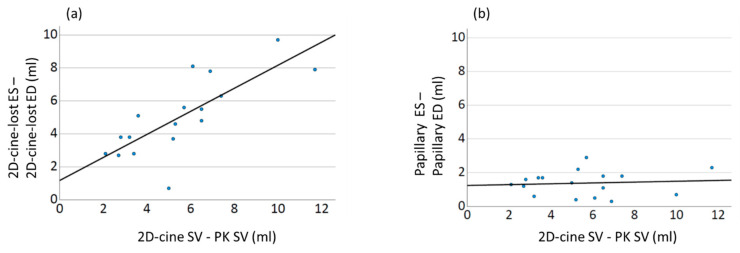
Correlation between 2D-cine volume discrepancies of edge structures/papillary muscle in end-systole and end-diastole, and the deviation of 2D-cine SV (ESV) from the gold standard PK measurements. The solid line in the plots represents the linear regression line. (**a**) Strong correlation (R2 0.71, *p* < 0.01) is observed between the volume discrepancies of 2D-cine residual structures (2D-cine-lost) in ES and ED, and the deviation of 2D-cine calculated SV from PK measurements. (**b**) In contrast, no correlation (R2 0.15, *p* < 0.35) is found between the volume discrepancies of papillary muscles (Papillary) in ES and ED, and the deviation of 2D-cine calculated SV from PK measurements.

**Table 1 diagnostics-13-03162-t001:** Quality and Time Requirements of 2D-Cine and 3D-wh methods.

		2D-Cine Mean (±SD)	3D-wh Mean (±SD)	*p*-Value
Image Quality	CNR	62.5 (9.7)	41.8 (15.7)	<0.01
	SNR	89.9 (12.3)	88.8 (21.6)	0.13
Duration (min)	Post-Processing	5.6 (1.1)	8.1 (2.3)	<0.01
	Acquisition	3.5 (1.6)	11.5 (7.8)	<0.01

The contrast-to-noise ratio (CNR) and signal-to-noise ratio (SNR) were determined via regions of interest (ROIs) using the software syngo.via, Version 5.0.1, accessed on May 2022.

**Table 2 diagnostics-13-03162-t002:** Comparison of three-dimensional overlap between 2D-cine and 3D-wh volumes.

	ED Mean (±SD)	ES Mean (±SD)	*p-*Value
Dice coefficient	0.91 (±0.04)	0.89 (±0.03)	0.035
Hausdorff distance (mm)	1.31 (±0.56)	1.31 (±0.47)	<0.01
Volume Difference (mL)	2.3 (±1.13)	1.6 (±0.65)	<0.01

Two metrics were used: the Dice coefficient and Hausdorff distance. A high Dice’s coefficient value close to 1 indicates a high similarity and strong overlap between the surfaces, while a low value near 0 suggests low similarity or minimal overlap. The Hausdorff distance measures the average distance between surfaces in three-dimensional space and is reported in millimeters. Mean values with their respective standard deviations (SD) are provided. To assess the difference between end-systole (ES) and end-diastole (ED), a paired *t*-test was employed.

**Table 3 diagnostics-13-03162-t003:** Comparison of volumes at end-systole (ES) and end-diastole (ED) and the derived functional parameter EF from different MRI sequences.

	3D-wh Mean (±SD)	2D-cine Mean (±SD)	3D-wh vs. 2D-Cine *p*-Value	2D-Cine-adj Mean (±SD)	3D-wh vs. 2D-Cine-adj *p-*Value
ES (mL)	61.20 ± 11.4	50.60 ± 11.1	<0.01	59.60 ± 11.2	<0.01
ED (mL)	140.1 ± 16.2	135.0 ± 17.0	<0.01	137.8 ± 16.3	<0.01
EF (%)	56.42 ± 6.1	62.71 ± 5.6	<0.01	56.90 ± 5.9	0.04

The volumes from 3D-wh MRI sequences, 2D-cine MRI sequences, and the corrected volume 2D-cine-adj are compared. Mean values with their respective standard deviations (SD) are provided. The significance was assessed using paired *t*-tests.

**Table 4 diagnostics-13-03162-t004:** Stroke volume.

	SV (mL) Mean (±SD)	PC vs. *p*-Value
PC (SD)	78.8 ± 12.0	
2D-cine (SD)	84.4 ± 11.4	*p* < 0.01
2D-cine-adj (SD)	78.2 ± 12.0	*p* = 0.06
3D-wh (SD)	79.0 ± 12.1	*p* = 0.73

Presented are the mean values with standard deviations of the SV obtained from different volumetric methods (2D-cine, 3D-wh, 2D-cine-adj) and PC measurements. The gold standard PC was compared with each volumetric measurement using paired *t*-tests. Means with standard deviation (SD) are shown.

## Data Availability

The findings of this study are supported by data that can be made available upon reasonable request to the corresponding author. It’s important to note, however, that we cannot release the complete image datasets to the public due to the presence of personal details. Sharing these would violate the privacy and consent agreements with our study participants.

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
