# Peer review of "Comparative Study of 2D-Cine and 3D-wh Volumetry: Revealing Systemic Error of 2D-Cine Volumetry"

_diagnostics, 2023, doi:10.3390/diagnostics13203162_

Round 1

Reviewer 1 Report

Dear authors,

This was a great work! The study investigated whole heart 3D MRI with 2D MRI and found 3D to correlate well with another reference standard phase contrast 2D MRI. I have few comments and queries.

1. 3D whole heart MRI sequence- What is meant by respiratory control? Is this navigated? Free breathing? Pls elaborate on the acquisition. Since 2D is breath held and 3D probably free breathing- How much impact this can have on volume measurement?

2. Timing- 3D MRI takes 11 seconds is mentioned in text under results but 11  minutes in tables which seem more accurate. Pls correct that. Also, you can include that current third party commercial software applications could proivide automated measurment of 2d cine volumes in less than 30 seconds (for non congenital heart disease patients) and 3d whole heart requires manual measurement that is prone to variations. How is segmentation done for 3d mri? Is it first converted to short axis slices in Slicer? How is the data acquired? coronally/sagittal? 

3. PC measurement- Could overstimation of 2D SV compared to ascending aorta flow explained because of underestimation of PC flow at ascending aorta site. May prior studies have shown that flow at LVOT/aortic valve level correlates better to 2D cine volume compared to ascending aorta. PMID: 23857993; PMID: 27599727. Perhaps this can be discussed under limitations/ further research.

4. Pls include images of 3d and 2d mri for side to side comparsion of image quality.

Reviewer 2 Report

The authors compare 2D and 3D cardiac MRI protocols which are used to
extract functional parameters, such as the stroke volume. The
manuscript reads well and appears comprehensive. I am not an expert in
cardiac MRI and can not comment on the novelty of the manuscript.
Regardless, I have some comments for the authors to consider:    

- Acronyms are introduced at random. LV is only introduced near the
  end of the manuscript in line 343, EF twice in line 46 and 69 and ES
  even three times in line 317, 337 and 348. It would be preferred if
  they could be introduced once at their first occurrence consistently.

- The authors describe the 2D method as a set of 10 to 20 2D slices. I
  wonder if this is not actually 3D as a 3D model could be obtained by
  suitable rendering techniques? If, as the authors say, edge
  structures are not fully taken into account, more slices could be
  taken, potentially using a higher slice density at the edges.

- Comparing the imaging parameters for 2D-cine and 3D-whole-heart it
  seems that the main difference is the slice resolution, being an
  order of magnitude better for 3D-whole-heart. It should therefore
  not surprise anyone that the 3D method determines the volume more
  precisely. A fair comparison would use the same slice resolution
  (most likely returning very similar volumes) and benchmark the time
  required to make the measurement in each case.

- It appears that 2D-cine can be corrected very well to obtain correct
  numbers as demonstrated in Fig. 4 and 7. Considering the faster scan
  time I wonder if that suggests that 2D-cine is the method of choice.
  This is supported by the last paragraph in sec. 4.3.2. It is
  mentioned that correction methods are rarely used in clinical
  practise but that does mean that the 3D method is superior.

Author Response

Response to Reviewer 2 Comments

  1. Summary

Thank you very much for taking the time to review this manuscript. We appreciate your insights. Below, you will find our detailed responses to each comment. Additionally, we have highlighted the revisions and corrections in the re-submitted documents for easier reference.

  1. Point-by-point response to Comments and Suggestions for Authors

Comments 1: Acronyms are introduced at random. LV is only introduced near the end of the manuscript in line 343, EF twice in line 46 and 69 and ES even three times in line 317, 337 and 348. It would be preferred if they could be introduced once at their first occurrence consistently.

Response 1: Thank you for pointing out this inconsistency. We recognize the oversight regarding the use of acronyms, and we have reviewed the manuscript to ensure each acronym is introduced clearly and consistently upon its first occurrence.

Comments 2: The authors describe the 2D method as a set of 10 to 20 2D slices. I wonder if this is not actually 3D as a 3D model could be obtained by suitable rendering techniques? If, as the authors say, edge structures are not fully taken into account, more slices could be  taken, potentially using a higher slice density at the edges.

Response 2: Thank you for raising this point. It's important to clarify that the 2D-cine technique doesn't fundamentally represent a 3D visualization. Instead, it consists of individual layers assigned a specific thickness and is geometrically reassembled based on the cardiac cycle, meaning the layer acquisition occurs spatially independent of the heart's movement. During reconstruction, a cylinder representing primarily the left ventricle is derived from this data. While increasing the number of slices might enhance precision, it remains a reconstruction of the 3-dimensional ventricular content abstracted as a cylinder. In contrast, the 3D-wh sequence captures the entire interior of the ventricle up to the valves. This isn't a reconstruction of 3D reality from interpolated 2D images, but a genuine 3D capture of the LV. We have previously highlighted these points in the text (page number 2 paragraph 1. line 58-95). For better clarity, we've expanded on this topic in the introduction with the following addition „This removes the requirement for interpolation between various slices from different ac-quisitions, a process that is essential in 2D-cine MRI. Although enhancing the number of slices in 2D-cine MRI might refine the precision, the subsequently formed 3-dimensional volume still constitutes a cylindrical reconstruction and is especially susceptible to errors due to motion artifacts.“ (page number 2 paragraph 1. Line 95-100)

Comments 3: Comparing the imaging parameters for 2D-cine and 3D-whole-heart it   seems that the main difference is the slice resolution, being an order of magnitude better for 3D-whole-heart. It should therefore not surprise anyone that the 3D method determines the volume more precisely. A fair comparison would use the same slice resolution (most likely returning very similar volumes) and benchmark the time required to make the measurement in each case.

Response 3: Thank you for highlighting this concern. We're juxtaposing two fundamentally different acquisition techniques, both of which are tailored to represent a 3-dimensional heart volume. It's imperative to elucidate that the 2D-cine MRI method derives its data from a static spatial plane across a set duration. The 3D representation is then constructed by interpolating numerous adjacent planes, each attributed a virtual thickness. This resolution in the z-direction for the 2D-cine cannot be directly juxtaposed with the 3D-wh method, where an all-encompassing 3D capture of the heart is performed at a specific cardiac cycle point. What's pivotal is ensuring that the in-plane resolution and the image quality remain analogous for both methods, as this is instrumental for the precise demarcation of endocardial boundaries, which we've showcased in Table 1. Aiming for a similar slice thickness in 2D-cine as in 3D-wh would necessitate acquiring an order of magnitude more 2D-cine slices. But to what end?

Interestingly, our data shows that the broader layers in 2D-cine only introduce minimal interpolation effects. The 3D reconciliation of the volumes unveiled a substantial geometric congruence (refer to Table 2), and the derived volumes from the adjusted datasets suggest negligible variances between the two methods (see Table 3).

To emphasize this point further, we made explicit adjustments in the discussion section: „Interestingly, our study revealed that, despite the fundamental differences in image acquisition between the 2D-cine technique (with a significantly lower slice resolution) and the 3D whole-heart technique, there's a high overlap of the 3D volumetrics (table 2). Consequently, the derived comparative volume from both methods is almost identical (table 3).“ (page number 15 paragraph 4.3. line 506-510)

Comments 4:

It appears that 2D-cine can be corrected very well to obtain correct numbers as demonstrated in Fig. 4 and 7. Considering the faster scan time I wonder if that suggests that 2D-cine is the method of choice. This is supported by the last paragraph in sec. 4.3.2. It is mentioned that correction methods are rarely used in clinical practise but that does mean that the 3D method is superior.

Response 4:

Thank you for this insightful comment, which we definitely need to address. In our current study, we based our conclusions on the prevailing scenario where predominantly uncorrected cylindrical volumes are being used. Our research has been able to identify the primary cause of inaccuracies in volumetric measurements from 2D-cine slices, which, in theory, paves the way for possible corrections. Should the validation of this correction software prove successful in clinical routines, then the 2D-cine measurement would indeed be superior to 3D-wh. However, when considering the 2D-cine measurement technique currently in clinical use, 3D-wh seems more advantageous.

We have accordingly revised this in the manuscript:

„…The accuracy of these methods remains insufficiently verified. In conclusion, based on our study's findings, we believe that the application of 3D whole-heart volumetry is superior to the current clinical usage of 2D cine volumetry.“ (page number 17 paragraph 5. line 622-625)